# Improved Route to Linear Triblock Copolymers by Coupling with Glycidyl Ether-Activated Poly(ethylene oxide) Chains

**DOI:** 10.3390/polym15092128

**Published:** 2023-04-29

**Authors:** Daniel T. Krause, Susanna Krämer, Vassilios Siozios, Andreas J. Butzelaar, Martin Dulle, Beate Förster, Patrick Theato, Joachim Mayer, Martin Winter, Stephan Förster, Hans-Dieter Wiemhöfer, Mariano Grünebaum

**Affiliations:** 1Helmholtz Institute Münster, IEK-12, Forschungszentrum Jülich GmbH, Corrensstr. 46, 48149 Münster, Germany; 2MEET Battery Research Center, University of Münster, Corrensstr. 46, 48149 Münster, Germany; 3Karlsruhe Institute of Technology (KIT), Institute for Chemical Technology and Polymer Chemistry (ITCP), Engesserstraße 18, 76131 Karlsruhe, Germany; 4Jülich Centre for Neutron Science (JCNS-1/IBI-8), Forschungszentrum Jülich, Wilhelm-Johnen-Straße, 52425 Jülich, Germany; 5Ernst Ruska-Centre for Microscopy and Spectroscopy with Electrons, Physics of Nanoscale Systems (ER-C-1), Forschungszentrum Jülich, Wilhelm-Johnen-Straße, 52425 Jülich, Germany; 6Soft Matter Synthesis Laboratory, Institute for Biological Interfaces 3 (IBG-3), Karlsruhe Institute of Technology (KIT), Herrmann-von-Helmholtz-Platz 1, 76344 Eggenstein-Leopoldshafen, Germany; 7Ernst Ruska-Centre for Microscopy and Spectroscopy with Electrons, Materials Science and Technology (ER-C-2), Forschungszentrum Jülich GmbH, Wilhelm-Johnen-Straße, Jülich 52425, Germany; 8Jülich-Aachen Research Alliance, JARA, Fundamentals of Future Information Technology, Wilhelm-Johnen-Straße, 52425 Jülich, Germany; 9Institute of Physical Chemistry, RWTH Aachen University, Landoltweg 2, 52074 Aachen, Germany

**Keywords:** polymers, block copolymers, polyethylene oxide (PEO), convergent synthesis, epoxide, anionic polymerization, microphase separation, morphology

## Abstract

Poly(ethylene oxide) block copolymers (PEO_z_ BCP) have been demonstrated to exhibit remarkably high lithium ion (Li^+^) conductivity for Li^+^ batteries applications. For linear poly(isoprene)-*b*-poly(styrene)-*b*-poly(ethylene oxide) triblock copolymers (PI_x_PS_y_PEO_z_), a pronounced maximum ion conductivity was reported for short PEO_z_ molecular weights around 2 kg mol^−1^. To later enable a systematic exploration of the influence of the PI_x_ and PS_y_ block lengths and related morphologies on the ion conductivity, a synthetic method is needed where the short PEO_z_ block length can be kept constant, while the PI_x_ and PS_y_ block lengths could be systematically and independently varied. Here, we introduce a glycidyl ether route that allows covalent attachment of pre-synthesized glycidyl-end functionalized PEO_z_ chains to terminate PI_x_PS_y_ BCPs. The attachment proceeds to full conversion in a simplified and reproducible one-pot polymerization such that PI_x_PS_y_PEO_z_ with narrow chain length distribution and a fixed PEO_z_ block length of *z* = 1.9 kg mol^−1^ and a *Đ* = 1.03 are obtained. The successful quantitative end group modification of the PEO_z_ block was verified by nuclear magnetic resonance (NMR) spectroscopy, gel permeation chromatography (GPC) and differential scanning calorimetry (DSC). We demonstrate further that with a controlled casting process, ordered microphases with macroscopic long-range directional order can be fabricated, as demonstrated by small-angle X-ray scattering (SAXS), scanning electron microscopy (SEM) and transmission electron microscopy (TEM). It has already been shown in a patent, published by us, that BCPs from the synthesis method presented here exhibit comparable or even higher ionic conductivities than those previously published. Therefore, this PEO_z_ BCP system is ideally suitable to relate BCP morphology, order and orientation to macroscopic Li^+^ conductivity in Li^+^ batteries.

## 1. Introduction

As a polymer class, block copolymers (BCPs) are gaining continuous attention due to their remarkable properties such as an amphiphilic character or self-assembling ability [1,2,3]. For instance, an amphiphilic behavior enables in solution the formation of micelles, which are widely utilized in pharmaceutical applications, e.g., for drug delivery systems [1,2]. Moreover, their ability to self-assemble also in bulk paves the way to well-ordered morphologies, which find a wide range of applications [3,4,5], e.g., in lithography [6], semiconductor-based photocatalysis [4,7] and energy storage and conversion as fuel cell membranes [8], electrodes [9] or polymer electrolytes [10,11,12].

These aforementioned properties rely on the tailorable and unique structure of the respective BCP. The covalent binding of polar and nonpolar polymer blocks with defined block length could result in macromolecules with amphiphilic character and therefore tend to organize themselves into periodic, highly ordered, nano-sized domains, the so-called microphases [3,4,5,6,8,13,14,15,16]. The Flory–Huggins interaction parameter (χ) quantifies the incompatibility between the different blocks based on their interaction energy [13,14,15,16].

Our decision to choose the linear triblock copolymer poly(isoprene)-*b*-poly(styrene)-*b*-poly(ethylene oxide), denoted as PI_x_PS_y_PEO_z_, whereby each index (x, y and z) indicates the molar mass (*M*_n_) of the corresponding block in kg mol^−1^, was inspired by the work of Dörr et al. [4,7,8,9,10,11]. They applied a synthetic route developed in the group of Bates [17,18,19] and demonstrated its advantageous use as a template for detailed control of mesoscopic porous 3D architectures with embedded inorganic materials [4,7,8,9,10,11]. In our case, the synthesized PI_x_PS_y_PEO_z_ are supposed to be used as a structure-giving BCP matrix for conducting lithium ions (Li^+^). Additionally, Dörr and Pelz et al. showed that lowering the PEO_z_ chain length down to values around 45 repeating units (*M*_n_ ~2 kg mol^−1^) and with a low content (<4 vol.%) in relation to total BCP size resulted in a high Li^+^ conductivity [10,11].

Therefore, the PEO_z_ block has a significant influence on the Li^+^ transport properties and is directly linked to it, as the Li^+^ presumably only accumulate in this block, resulting in the formation of Li^+^ conducting pathways [10,11,20]. Consequently, those Li^+^ conducting pathways and, finally, the total Li^+^ transport are substantially determined by the properties of the PEO_z_ block and its resulting domain structure such as its size, long-range order, morphology and macroscopic orientation. This means for an optimal Li^+^ transfer, the structure-giving BCP matrix has to arrange itself into a long-range and highly ordered morphology, e.g., lamellar (LAM) or hexagonally close-packed cylindrical (HEX), continuously between two electrodes and an orientation connecting these electrodes. In order to obtain these properties in the PI_x_PS_y_PEO_z_, the combination of always having precisely defined length and a nearly monodisperse distribution in each block, as well as a controlled self-assembly during the membrane preparation process, is important [21,22].

Hence, in this work, the influence of the structure-giving BCP matrix on the short PEO_z_ chain order will be investigated. Corresponding PI_x_PS_y_PEO_z_ BCP will be synthesized by keeping the PEO_z_ chain length constant in order to vary its composition systematically and independently from each other (cf. Figure 1) [18,19,23,24]. For this purpose, PI_x_PS_y_PEO_z_ BCPs are varied in two different ways:

(1) By differing the ratio of the *M*_n_ of the PI_x_ to the PS_y_ block (*M*_n,PIx_/*M*_n,PSy_), while retaining the same PEO_z_ block proportion because the total *M*_n_ of the BCP (*M*_n,total_) is hold constant (cf. Figure 1a).

(2) By altering the *M*_n,total_ and therefore the PEO_z_ block proportion, while keeping *M*_n,PIx_/*M*_n,PSy_ = constant (cf. Figure 1b).

This will be achieved by a new developed synthesis route, which ensures the use of consistent identical and very short as well as commercially available prefabricated PEO_z_ chains for the attachment to BCPs.

Usually, such BCPs are prepared in a stepwise manner by synthesizing each polymer block sequentially [17,18,19,25]. Thus, each polymer block is formed from the respective monomers in a series of polymerization steps according to the principle of living sequential anionic polymerization (cf. Figure 2I), as it offers the highest control over the polymerization process, hence the dispersity (*Đ*), and also proceeds without side reactions [5,17,18,19,26,27,28,29]. However, due to the explosive and highly toxic properties of ethylene oxide (EO) gas, its use implies special safety requirements [30]. Therefore, the use of a short prefabricated PEO_z_ block in our synthesis route leads to the fact that the handling of EO gas monomers during PI_x_PS_y_PEO_z_ polymerization can be avoided. In this way, the necessary use of EO gas for the synthesis of PEO_z_ chains can be carried out in a separate and upstream synthesis step.

Thus, in this study, an exact defined methoxy PEO (mPEO_z_) chain with a modified end group was chosen, enabling it accessible for direct and covalent attachment to the stable PI_x_PS_y_^−^ carbanion of the living polymer chain. Considering the large variety of suitable end groups, tethering an epoxide end group to the mPEO_z_ chain (EmPEO_z_) enables a selective single one-step addition to the PI_x_PS_y_^−^ anion by utilization of the strong Li-O interaction (cf. Figure 2II) [5,17,31,32,33,34]. This is similar to the general strategy of using epoxides as terminating agents as reported in literature [35,36,37,38,39,40,41]. The strong interaction between the hard oxygen anion and the hard Li^+^ can be well explained based on the concept of “hard” and “soft” acids and bases (HSAB) [32,42]. In addition to the PEO_z_ chain, at the junction point only an extra alcohol group is introduced in the polymer as (poly(isoprene)-*b*-poly(styrene)-*b*-alcohol methoxy poly(ethylene oxide) = PI_x_PS_y_AmPEO_z_).

Our convergent synthetic method based on the modular principle aims to create access to PI_x_PS_y_PEO_z_ with constantlythe same very short and well-defined PEO_z_ block, allowing the PI_x_ and PS_y_ blocks to be varied systematically and independently. Moreover, as this synthesis route only utilizes commercially available chemicals, it offers a high reproducibility and up-scaling probability, making tailored and precisely defined PI_x_PS_y_AmPEO_z_ BCPs accessible for large-scale production. We have already published a prior patent application for this synthesis method [43].

## 2. Materials and Methods

### 2.1. Materials

Sodium *tert*-butoxide (NaO*^t^*Bu, 99.9%, Sigma-Aldrich, Darmstadt, Germany) was purified by sublimation [44] (105 °C at ≤3 × 10^−3^ mbar), 3 Å molecular sieves (VWR, Darmstadt, Germany) was activated by drying at 300 °C under vacuum < 1 × 10^−6^ mbar and methoxy poly(ethylene oxide) (mPEO_1.9_ equals to *M*_n_ = 1.9 kg mol^−1^, VWR) was dried at 30 °C under vacuum < 1 × 10^−6^ mbar and all were subsequently stored inside a glovebox (MBraun Unilab, Garching, Germany, ≤0.1 ppm of Water (H_2_O) and oxygen (O_2_)) under argon atmosphere. Epichlorohydrin (≥99.0%, Sigma-Aldrich), chloroform-*d*_1_ (CDCl_3_, 99.8% D, VWR) and toluene (≥99.85%, VWR) were dried using activated 3 Å molecular sieves until ≤1 ppm H_2_O, tetrahydrofuran (THF, ≥99.8%, unstab., Alfa Aesar, Kandel, Germany) was dried using activated 3 Å molecular sieves until ≤5 ppm H_2_O; all were subsequently stored inside a glovebox under argon atmosphere and passed through syringe filter (polytetrafluoroethylene (PTFE) membrane, pore size = 0.2 µm, VWR) prior to use. Seven days before use, isoprene (≥99%, VWR) and styrene (≥99%, Sigma-Aldrich) were dried (still stored in glovebox fridge) using activated 3 Å molecular sieves until ≤1 ppm H_2_O and were distilled under vacuum directly before use. A total of 1.4 M *sec*-butyllithium solution in cyclohexane (*sec*-BuLi, Sigma-Aldrich) was used as received. The concentration of *sec*-BuLi was directly determined by double titration, using a glass-coated magnetic stir bar and the ready-to-use reagent: 2-propanol solution in toluene with 0.2% 1,10 phenanthroline indicator titration solution for quantitative analysis of butyllithium (Sigma-Aldrich), prior to use. Diethyl ether (Et_2_O, ≥99.9%, inhibitor-free, Sigma-Aldrich), dichloromethane (DCM, ≥99.5%, VWR), methanol (MeOH, ≥99.9%, VWR) and 0.5 M hydrogen chloride solution in MeOH (Sigma-Aldrich) were used as received. For quantitative water content determination, a Karl Fischer coulometric titrator C30S (Mettler-Toledo, Gießen, Germany) with a platinum generator electrode without a diaphragm was used.

### 2.2. End Group Modification (EmPEO_1.9_)

The complete end group modification reaction of mPEO_1.9_ to EmPEO_1.9_ was carried out at room temperature under argon atmosphere in a glovebox. NaO*^t^*Bu (1.5 equiv, 9.32 mmol, 0.896 g) was dissolved in THF (60 mL) and subsequently added to a solution of mPEO_1.9_ (1.0 equiv, 6.29 mmol, 12.0 g) in THF (60 mL). After 72 h, epichlorohydrin (8.0 equiv, 50.3 mmol, 4.65 g, 3.94 mL) was added dropwise to the reaction within 15 min and stirred for six days. Subsequently, the volatile components were removed at 55 °C under vacuum and the solid residue was dissolved in THF. Undissolved components, mainly formed sodium chloride (NaCl), were removed by centrifugation (Sigma 3-18KS, Osterode am Harz, Germany, 10000 rpm for 10 min), followed by filtration with syringe filter (PTFE membrane, pore size = 0.2 µm, VWR) and drying at room temperature under vacuum. Afterwards, the product was dissolved in a little amount of toluene at 40 °C, precipitated into cold Et_2_O and collected by centrifugation as before. This process was repeated three times. The resulting EmPEO_1.9_ was dried at 30 °C under vacuum < 1 × 10^−6^ mbar (yield: 89–93%). The product was characterized by ^1^H-nuclear magnetic resonance (NMR) spectroscopy, ^13^C-NMR spectroscopy, gel permeation chromatography (GPC), differential scanning calorimetry (DSC) and thermogravimetric analysis (TGA).

### 2.3. Synthesis of Poly(isoprene)-b-poly(styrene)-b-alcohol Methoxy Poly(ethylene oxide) (PI_x_PS_y_AmPEO_1.9_)

The complete polymer synthesis reaction of PI_x_PS_y_AmPEO_1.9_ was carried out at room temperature under argon atmosphere in a glovebox using a glass-coated magnetic stir bar for mixing. The PI_x_ and PS_y_ block was synthesized by living sequential anionic polymerization as reported in the literature (Figure 2I) [17,18,19]. All PI_x_PS_y_AmPEO_1.9_ materials were synthesized, as exemplarily described for the PI_14.6_PS_34.8_AmPEO_1.9_ in the following, where only the *sec*-BuLi, monomers (isoprene and styrene) and EmPEO_1.9_ amounts were adjusted according to the desired composition (cf. Table 1).

For preparation of the PI_14.6_ block, isoprene (214 equiv, 53.1 mmol, 3.62 g) was dissolved in toluene (120 mL) followed by the addition of 1.4 M *sec*-BuLi solution in cyclohexane (1.0 equiv, 0.249 mmol, 178 µL). After stirring for 24 h, to ensure a quantitative conversion of isoprene monomers [45], styrene (334 equiv, 83.1 mmol, 8.66 g) was added to the yellowish reaction solution of the living PI_14.6_^−^ anion to build the PS_34.8_ block. After stirring for 24 h, to ensure a quantitative conversion of styrene monomers [45], EmPEO_1.9_ (1.1 equiv, 0.274 mmol, 0.521 g) was added to the reddish reaction solution of the living PI_14.6_PS_34.8_^−^ anion to attach the PEO_1.9_ block. After stirring for 48 h, 0.5 M hydrogen chloride solution in MeOH (1.5 equiv, 0.37 mmol, 497 µL) was added to the colorless reaction solution. Subsequently, the volatile components were removed at 40 °C under vacuum. Then, the product was dissolved in DCM, precipitated into MeOH and this process was repeated three times. The resulting polymer was dried at 50 °C under vacuum < 1 × 10^−6^ mbar (yield: 92–95%). The product was obtained as a white solid and characterized by ^1^H-NMR spectroscopy, GPC, DSC and TGA.

### 2.4. Sample Preparation of PI_x_PS_y_AmPEO_1.9_ Membranes for Morphological Characterization

The complete PI_x_PS_y_AmPEO_1.9_ membrane casting process, for preparing the samples for the morphological characterization, was performed under argon atmosphere. In a glovebox, an 8 wt.% solution of the dried PI_x_PS_y_AmPEO_1.9_ in THF was prepared and followed by transferring the mixture into a PTFE crucible. The filled PTFE crucible was placed in a Schlenk vessel and connected to the argon of a Schlenk line. A very low and constant argon flow over 6 days at room temperature in a THF-saturated atmosphere was used to allow a controlled evaporation to achieve distinct microphase separation.

Subsequently, for small-angle X-ray scattering (SAXS) measurement, the resulting membrane (cf. Appendix A) was carefully broken into smaller pieces in order to fit the sample into the glass capillary (borosilicate glass, outer diameter = 2.1 mm, wall thickness = 0.05 mm, Hilgenberg, Malsfeld, Germany). The filled glass capillary was sealed tightly.

For scanning electron microscopy (SEM) and transmission electron microscopy (TEM) investigations, a small piece of the resulting membrane (cf. Appendix A) was cut into ultrathin sections of about 50–100 nm using a cryo-ultramicrotome (cf. Appendix A).

### 2.5. Nuclear Magnetic Resonance (NMR) Spectroscopy

The NMR spectra of PI_14.6_PS_34.8_AmPEO_1.9_, PI_35.1_PS_14.8_AmPEO_1.9_ and EmPEO_1.9_ were recorded using an AVANCE NEO 500 MHz (Bruker, Billerica, MA, USA), that of mPEO_1.9_, PI_24.8_PS_25.0_AmPEO_1.9_ and PI_26.1_PS_67.3_AmPEO_1.9_ using an AVANCE NEO 400 MHz (Bruker) and for PI_6.8_PS_17.3_AmPEO_1.9_ using an AVANCE III 400 MHz (Bruker). NMR measurements with number of scans = 64 were recorded, and the recycle delay *D*1 between transients was set to 30 s to ensure full relaxation to equilibrium magnetization and thus the acquisition of quantitative spectra, except for mPEO_1.9_. All NMR spectra were recorded at 300 K using CDCl_3_ as deuterated solvent, where all signals were referenced to CDCl_3_ (δ = 7.3 ppm for ^1^H and δ = 77.2 ppm for ^13^C relative to tetramethylsilane) [46]. The spectra were analyzed with the software MestReNova (version: 12.0.4-22023, Mestrelab Research, Santiago de Compostela, Spain).

### 2.6. Gel Permeation Chromatography (GPC)

GPC was carried out in THF using an HLC-8320GPC EcoSEC (Tosoh Bioscience, Griesheim, Germany) system equipped with three PSS SDV columns (100, 1000, 100,000 Å) (8 × 300 mm) of 5 μm, a UV and a differential refractive index (RI) detector. The operation temperature was set to 35 °C with a flow rate of 1 mL min^−1^. Calibration of the system was carried out with poly(styrene) standards ranging from 800 to 2.2 × 10^6^ g mol^−1^. Typically, 50 μL of a 2.0 mg mL^−1^ sample solution was injected onto the columns.

### 2.7. Differential Scanning Calorimetry (DSC)

DSC was conducted using a heat flux calorimeter DSC-Q2000 (TA Instruments, New Castle, DE, USA) with LNCS (Liquid Nitrogen Cooling System) and the Tzero^®^-technology for the precise recording of the baseline. Under argon atmosphere (inside glovebox), ~10 mg of sample was enclosed in hermetically sealed Tzero^®^ aluminum pans. Two heating ramps in the temperature range from −140 °C to 190 °C with a heating rate of 10 K min^−1^ under helium as sample purge (25 mL min^−1^) were measured for all samples. The DSC signals were analyzed with the Universal Analysis 2000 software (version: 4.5A, Build 4.5.0.5, TA Instruments).

### 2.8. Thermogravimetric Analysis (TGA)

For TGA, an aluminum pan was filled with ~10 mg of sample and hermitically sealed under argon atmosphere in a glovebox, subsequently loaded to the device without contact to ambient air and pierced in the furnace under helium atmosphere. The measurement was carried out on a TGA-5500 with IR furnace (TA Instruments) under helium flow (25 mL min^−1^) with a constant heating rate of 2 K min^−1^ from 30 °C to 600 °C. The TGA signals were analyzed with the TA Instruments TRIOS software (version: 5.1.1.46572, TA Instruments).

### 2.9. Small-Angle X-ray Scattering (SAXS)

The instruments “Ganesha-Air” from SAXSLAB/XENOCS (Grenoble, France) and Gallium Anode Low-Angle X-ray Instrument (GALAXI) were used. The X-ray source of the laboratory-based “Ganesha-Air” system is a D2-MetalJet (Excillum, Kista, Sweden) with a liquid metal anode operating at 70 kV and 3.57 mA with Ga–K*α* radiation (wavelength λ = 0.134 nm). The beam is further focused with a focal length of 55 cm, using especially made X-ray optics (Xenocs) to provide a very narrow and intense beam at the sample position. Two pairs of scatterless slits are used to adjust the beam size depending on the detector distance. The data were acquired with a position-sensitive detector (PILATUS 300 K, Dectris, Baden-Daettwil, Switzerland). After calibration with silver behenate, the distance from the sample to the detector was set to 950 and 350 mm resulting in a *Q*-range 0.13–6.00 nm^−1^. All samples were sealed in glass capillaries of 2 mm inner diameter. Data reduction and background subtraction were performed using the Python-based project Jscatter [47]. Fitting of radially averaged SAXS curves was done using Scatter [48].

### 2.10. Cryo-Ultramicrotomy

To prepare ultra-thin sections for electron microscopy, a Leica EM UC7 ultramicrotome (Wetzlar, Germany) equipped with an EM FC7 cryo-chamber was used. Temperature of sample, knives and chamber was set to −80 °C. The samples were trimmed with a diamond trimming knife from Diatome (trim 45, Nidau, Switzerland), and ultra-thin sections were made with a cryo-immuno diamond knife also from Diatome (cf. Appendix A). Ultra-thin sections were collected dry on carbon-coated copper grids and section thickness was set to 50 nm.

### 2.11. Scanning Electron Microscope (SEM)

SEM measurements were performed using a Thermo-Fisher Volumescope (Waltham, MA, USA). Images were taken on unstained samples in high vacuum at an accelerating voltage of 30 kV and a working distance of 10 mm using an annular ring STEM detector in bright field mode at room temperature.

### 2.12. Transmission Electron Microscope (TEM)

For TEM measurements, a JEOL JEM-F200 (Freising, Germany) with field emission gun (FEG) operating at an accelerating voltage of 200 kV was used. Images were taken of the unstained sample with a STEM bright field detector at room temperature.

## 3. Results and Discussion

### 3.1. Strategy

In line with the introductory part, the main focus of this work is to develop a convergent synthesis route which enables the independent and systematic PI_x_ and PS_y_ block variation in BCPs with constantly the same PEO_z_ block size, whereby the complete control over morphology and orientation of the BCP, especially of the very short PEO_z_ block, should be obtained, as this is crucial for its possible application. For this purpose, PEO_z_ blocks with an identical chain length were chosen and introduced into PI_x_PS_y_PEO_z_, in which the PI_x_ and PS_y_ blocks are synthesized by anionic polymerization of the monomers and therefore easily and precisely modified.

To do so, the commercially available and prefabricated mPEO_z_ (*M*_n_ = 1.9 kg mol^−1^) was activated for the attachment by functionalization with an epoxide end group (yielding EmPEO_z_). In our synthesis route, the EmPEO_z_ can be attached and thus covalently linked to the previously anionically synthesized PI_x_PS_y_^−^ carbanion by a simple one-step addition (cf. Figure 2). Herein, it is important that an appropriate end group is selected, because it will have a great influence on the properties of the whole PEO_z_ block of the BCP, especially in our focus of very short PEO_z_ chains [49]. Thus, the use of an epoxide as an end group ensures that the formed junction point in PI_x_PS_y_AmPEO_z_ hardly differs from the PI_x_PS_y_PEO_z_ obtained by using EO gas, i.e., only by an additional OH-group. This smallest possible difference enables the selective consideration solely of the variation of the two nonpolar blocks between the different synthesized PI_x_PS_y_AmPEO_z_. There are approaches that have been reported in literature thus far in which, for instance, benzophenone or diphenylethylene were attached to the PEO_z_ chain as terminal groups [49]. However, regarding the reported end group modifications, multi-step synthesis has to be employed, making the overall synthesis of a PI_x_PS_y_PEO_z_ more complex, and these groups are more chemically different compared to the ether and OH-group with unintended influence on the polarity [50]. Especially in the case of short PEO_z_ chains, this could have a great impact on its domain structure formation.

Considering the end group modification shown in Figure 3, the PEO_z_ block was end-capped with an epoxide group in a simple literature-known and modified one-pot reaction [51,52]. In the first modification step, the terminal OH-group of mPEO_z_ is selectively deprotonated by a strong base, i.e., NaO*^t^*Bu. Afterwards, the formed mPEO_z_ alcoholate attacks the epoxide group of the added epichlorohydrin via a nucleophilic (S_N_2) attack to form the desired product EmPEO_z_ (Figure 3). This method can also be extended for longer (*M*_n_ = 6 kg mol^−1^) PEO_z_ chains as reported by van Butsele et al. [52]. Here it should be mentioned, that the PEO_z_ block has to be selectively functionalized at only one terminal OH-group. End-capping both terminal OH-groups would lead to the undesired formation of a symmetric five BCP (PI_x_PS_y_PEO_z_PS_y_PI_x_). As mPEO_z_ only possess one terminal OH-group allowing for a distinct functionalization, the linkage to the PI_x_PS_y_^−^ anion happens in a selective manner, as indicated in (Figure 3).

Another important aspect of the above-described usage of the prefabricated mPEO_z_ block is the accurate characterization prior to its linkage, because a PEO_z_ block that is already attached to the PI_x_PS_y_ block is relatively small compared to the total BCP and therefore difficult to characterize and control accurately, especially in terms of its *Đ*.

The straightforward addition of EmPEO_z_ to the reaction solution furthermore simplifies the polymerization by terminating the living PI_x_PS_y_^−^ anion because it allows a one-step attachment in the presence of the Li^+^ counterion in nonpolar solvent (Figure 2II). After the addition of the EmPEO_z_ to the reaction solution, the PI_x_PS_y_^−^ anion attacks the epoxy group of the EmPEO_z_ via a ring-opening reaction forming an alcoholate group and a covalent bond between the PI_x_PS_y_ block and the AmPEO_z_. Hence, the charge of the carbanion is transferred to the O-atom of the alcoholate group. The negative charge on the O-atom of the alcoholate group is directly blocked by the Li^+^ counterion, which strongly reduces the nucleophilicity. Therefore, a new ring-opening reaction with another epoxide group of a second EmPEO_z_ molecule is inhibited due to the strong O-Li-ion pair (Appendix A), thus blocking a further polymerization, or any other undesired crosslinking or side reaction and highlighting the advantage of this one-pot synthesis route (Figure 2) [17,31,37,38,39,40]. Especially, a reaction of the living anions with O_2_ is suppressed, so that immediately after the AmPEO_z_ block attachment, the product is stable in air, providing an advantage in terms of reproducibility and up-scalability [53,54,55].

Moreover, the epoxide functionalization allows the EmPEO_z_ to be added to the reaction solution in minimal excess (cf. Table 1), so that the PI_x_PS_y_^−^ anions react stoichiometrically to form the PI_x_PS_y_AmPEO_z_ and the excess and unbound EmPEO_1.9_ is easily washed out after the polymerization reaction.

### 3.2. Synthesis

Triblock copolymers were synthesized in one-pot polymerization in three subsequential steps by living anionic polymerization. As indicated by Figure 2Ia, the PI_x_ block was prepared from isoprene monomers using *sec*-BuLi serving as initiator for formation of the living carbanion. Subsequently, the living PI_x_^−^ anion was used for chain extension by addition of styrene (Figure 2Ib). In the third step of this synthesis route, the PEO_z_ block was covalently attached to the living PI_x_PS_y_^−^ anion via a one-step reaction of the epoxide group-functionalized EmPEO_z_ and thus terminated the polymerization to the BCP (Figure 2II). The coupling of EmPEO_z_ to a PI_x_PS_y_^−^ anion enabled the selective and highly controllable preparation of PI_x_PS_y_AmPEO_z_ in which the PEO_1.9_ block features an identical chain length and a *Đ* very close one in all cases. The respective *M*_n,PIx_/*M*_n,PSy_ and thus the *M*_n,total_ of the BCP can be specifically tailored by systematically varying the used amount of initiator and the PI_x_ and PS_y_ monomers. However, as the living anionic polymerization is highly sensitive to impurities, because they directly lead to the termination of the corresponding living anion, special requirements for the synthesis conditions must always be ensured (cf. experimental section).

Here, it is worth noticing that the reaction of mPEO_z_ to EmPEO_z_ has to be quantitative, as no separation of both can be conducted due to the strong chemical similarity of mPEO_z_ and EmPEO_z_. Even more, since acidic protons of residual mPEO_z_ would protonate the PI_x_PS_y_^−^ anions upon addition and therefore inhibit the reaction of EmPEO_z_ and the PI_x_PS_y_^−^ anions, the importance of a quantitative functionalization reaction is emphasized. In order to obtain a quantitative end group modification of mPEO_z_, the base used for deprotonation of the OH-group has to be a weak nucleophile, as otherwise it would compete with the formed mPEO_z_ alcoholate regarding the S_N_2 reaction with epichlorohydrin, thus minimizing the yield of the desired EmPEO_z_. Additionally, the following conditions have been optimized for the quantitative formation of EmPEO_z_: (1) The use of NaO*^t^*Bu and epichlorohydrin in excess (relative to the mPEO_z_). (2) The reaction has to be carried out under argon atmosphere using anhydrous reactants to ensure that the precipitation of the formed NaCl in THF is quantitatively [56]. Furthermore, the absence of H_2_O prohibits the formation of nucleophilic OH^−^-ions, which would also compete with the formed mPEO_z_ alcoholate regarding the reaction with epichlorohydrin.

By the use of EmPEO_1.9_, different BCPs with constant PEO_1.9_ block size were synthesized, using this universally applicable and simplified synthesis method, by systematically varying the *M*_n_ of their PI_x_ and PS_y_ blocks (cf. Figure 1), namely PI_6.8_PS_17.3_AmPEO_1.9_, PI_14.6_PS_34.8_AmPEO_1.9_, PI_24.8_PS_25.0_AmPEO_1.9_, PI_35.1_PS_14.8_AmPEO_1.9_ and PI_26.1_PS_67.3_AmPEO_1.9_ (cf. Figure 4). While the *M*_n,total_ of the BCP and thus the PEO_1,9_ block fraction remained constant, the *M*_n,PIx_/*M*_n,PSy_ was varied (cf. Figure 1a). According to this, for PI_24.8_PS_25.0_AmPEO_1.9_ the *M*_n,PIx_ is equal to the *M*_n,PSy_, in the case of PI_35.1_PS_14.8_AmPEO_1.9_ the *M*_n,PIx_ is doubled compared to the *M*_n,PSy_, whereas for PI_14.6_PS_34.8_AmPEO_1.9_ the *M*_n,PSy_ is two times larger than the *M*_n,PIx_, which is exactly the opposite compared to PI_35.1_PS_14.8_AmPEO_1.9_.

In addition, by keeping a *M*_n,PIx_/*M*_n,PSy_ constant, in which the *M*_n,PSy_ is doubled compared to the *M*_n,PIx_, the PEO_1.9_ block proportion was varied by changing the *M*_n,total_ of the BCP (cf. Figure 1b). Therefore, by halving the *M*_n,total_ of PI_14.6_PS_34.8_AmPEO_1.9_, the PEO_1.9_ block fraction was doubled in PI_6.8_PS_17.3_AmPEO_1.9_. In contrast, for PI_26.1_PS_67.3_AmPEO_1.9_ the *M*_n,total_ was varied exactly in the opposite way, i.e., the *M*_n,total_ of PI_14.6_PS_34.8_AmPEO_1.9_ was doubled, resulting in a reduced PEO_1.9_ block proportion.

### 3.3. NMR Characterization

#### 3.3.1. EmPEO_1.9_

First, the EmPEO_1.9_ obtained from the functionalization reaction of mPEO_1.9_ (Figure 3) was analyzed for its quality via ^1^H- and ^13^C-NMR to verify that the end group modification was quantitative as required. The ^1^H-NMR spectrum of EmPEO_1.9_ (Figure 1) shows the proton and satellite peaks (highlighted by the grey box) from the polyether chain in the chemical shift region of δ = 3.4–3.7 ppm (c in orange circle) [51,52]. The sharp singlet at δ = 3.3 ppm (d in grey circle) can be assigned to the three protons of the terminal methoxy group [51,52]. The three protons of the epoxide group attached by functionalization reaction split into three characteristic signals located at δ = 2.5 ppm (a in blue circle), δ = 2.7 ppm (a’ in blue circle) and δ = 3.1 ppm (b in yellow circle) [51,52]. The integrals of these three signals are equal and have a value of one with respect to the three protons of the terminal methoxy group, indicating that the attachment of the epoxide group took place quantitatively.

The comparison of the ^13^C-NMR spectra of mPEO_1.9_ and EmPEO_1.9_ (Figure 2) highlights the characteristic carbon signal changes due to the modification reaction (cf. for entire spectrum of mPEO_1.9_ Appendix A and EmPEO_1.9_ Appendix A). For mPEO_1.9_ (Figure 2a), the carbon atom signal at δ = 61.7 ppm (a in dark blue circle) corresponds to the carbon atom directly linked to the OH-group and the signal at δ = 72.5 ppm to its directly adjacent carbon atom (b in yellow circle). Both carbon atom signals are not detected in the ^13^C-NMR spectrum of the EmPEO_1.9_ (Figure 2b), confirming that the functionalization of all OH-groups with epoxide groups to form EmPEO_1.9_ is quantitatively [51]. The carbon atoms of the polyether chain at δ = 70.5 ppm (c in orange circle) and the terminal methoxy group at δ = 59.0 ppm (e in light green circle) are not affected during the modification reaction, thus the respective signals can be found in both ^13^C-NMR spectra [51]. In contrast to the ^13^C-NMR spectrum of mPEO_1.9_ (Figure 2a), two carbon atom signals appear at δ = 44.1 ppm (f in light blue circle) and δ = 50.7 ppm (g in brown circle) in ^13^C-NMR spectrum of EmPEO_1.9_ (Figure 2b) which are characteristic for the attached epoxide group [51].

The successful and quantitative attachment of the epoxide group to the mPEO_1.9_ molecule was verified by the integrals in ^1^H-NMR as well as the altered signals in ^13^C-NMR. Furthermore, the absence of additional signals in the ^1^H- and ^13^C-NMR spectra of EmPEO_1.9_ (Figure 1 and Figure 2b) indicated that during the modification reaction, no side reaction occurred, and no impurities were introduced (cf. Appendix A for entire spectrum).

#### 3.3.2. PI_x_PS_y_AmPEO_1.9_

The quantitatively epoxy-functionalized EmPEO_1.9_ was used for the synthesis of different PI_x_PS_y_AmPEO_1.9_, which were subsequently characterized by ^1^H-NMR measurements. In the following, PI_14.6_PS_34.8_AmPEO_1.9_ is discussed exemplarily based on its ^1^H-NMR result, whereas in Appendix A and Appendix A, the results of all synthesized PI_x_PS_y_AmPEO_1.9_ are summarized, while the individual spectra are depicted in Appendix A.

The ^1^H-NMR spectrum of PI_14.6_PS_34.8_AmPEO_1.9_ is displayed in Figure 3. The signals in the range of δ = 7.4–6.2 ppm can be assigned to the five aromatic protons of the phenyl group of PS_34.8_ block (red circles). The signal of the olefinic proton from the 1,4-PI_14.6_ block (blue circle) is located at δ = 5.2 ppm and the two terminal protons of the 3,4-PI_14.6_ block (blue circle) are located at δ = 4.8 and 4.7 ppm, respectively [57]. The integral ratio of the 1,4-PI_14.6_ block to the 3,4-PI_14.6_ block is 1.00: 0.15, being the same for all prepared PI_x_PS_y_AmPEO_1.9_ [57]. The protons of the polyether chain of the AmPEO_1.9_ (dark green circle) are localized at δ = 3.7 ppm. The signal at δ = 3.4 ppm can be attributed to the terminal methoxy group of AmPEO_1.9_ (light green circle), and the alkyl backbone protons of the entire PI_14.6_PS_34.8_AmPEO_1.9_ (light grey circles) are located in the range of δ = 2.3–1.3 ppm. The fact that the proton signals from AmPEO_1.9_ are still present after the purification procedure of PI_14.6_PS_34.8_AmPEO_1.9_ indicates that AmPEO_1.9_ was covalently linked to the PI_14.6_PS_34.8_^−^ anion by a nucleophilic attack on the epoxide group of EmPEO_1.9_, as described previously (cf. Figure 2II). The nonbonded EmPEO_1.9_ was removed during the purification procedure because it dissolves in MeOH, which is used in the purification process [58].

The theoretical molar mass (*M*_n,calc._) of the individual blocks shown in Table 2, which was calculated from the ratio of the used masses of monomers (isoprene and styrene) to *sec*-BuLi (initiator) (cf. Table 1), fit to the respective ^1^H-NMR integrals of the characteristic protons from the PI_x_, PS_y_ and AmPEO_1.9_ blocks for all synthesized PI_x_PS_y_AmPEO_1.9_ (Table 3, Appendix A, Appendix A).

Considering the fact that the molar masses (*M*_n,NMR_) of the individual blocks shown in Table 3 determined from the respective proton number of ^1^H-NMR integrals (cf. Appendix A) agree with the corresponding *M*_n,calc._ indicates a nearly stoichiometric conversion of the utilized masses listed in Table 1 to the desired PI_x_PS_y_AmPEO_1.9_ without any side reactions. This close to quantitative polymerization process is due to the high reaction control by living anionic polymerization, the controlled synthesis conditions and the use of a slight excess of the EmPEO_1.9_ during the polymer synthesis. The slight excess (10%) of EmPEO_1.9_ ensures that all PI_x_PS_y_^−^ anions are stoichiometrically saturated with the corresponding AmPEO_1.9_ blocks, whereby the chosen epoxide end group modification ensures that not more than one EmPEO_1.9_ molecule can be covalently attached per PI_x_PS_y_^−^ anion (cf. Figure 2 and Appendix A). Therefore, the excess of unattached EmPEO_1.9_ was removed during the purification process. In addition, the strong O-Li-ion pair acts as a kind of protecting group, avoiding unwanted crosslinking, side and termination reactions.

### 3.4. GPC Measurements

The GPC measurements were conducted to determine the molar mass (*M*_n,GPC_), to confirm the *M*_n,NMR_ results, the efficiency of the polymer synthesis and to determine the *Đ* of the EmPEO_1.9_ as well as the synthesized PI_x_PS_y_AmPEO_1.9_. The corresponding GPC traces are shown in Figure 4, and Table 4 lists the results (cf. Appendix A for the complete traces). From the GPC traces, it can be seen that all synthesized PI_x_PS_y_AmPEO_1.9_ polymers and EmPEO_1.9_ have a narrow, unimodal shape and as a result, a *Đ* smaller than 1.10. These are: PI_6.8_PS_17.3_AmPEO_1.9_ (*Đ* = 1.02), PI_14.6_PS_34.8_AmPEO_1.9_ (*Đ* = 1.02), PI_24.8_PS_25.0_AmPEO_1.9_ (*Đ* = 1.03), PI_35.1_PS_14.8_AmPEO_1.9_ (*Đ* = 1.03), PI_26.1_PS_67.3_AmPEO_1.9_ (*Đ* = 1.08) with an additional minor signal, and in comparison, EmPEO_1.9_ (*Đ* = 1.03). Moreover, it can be clearly observed by the overlapping of the traces that the *M*_n,GPC_ of PI_14.6_PS_34.8_AmPEO_1.9_, PI_24.8_PS_25.0_AmPEO_1.9_ and PI_35.1_PS_14.8_AmPEO_1.9_ are almost identical and in the range of *M*_n,GPC_ = 62.4–65.3 kg mol^−1^, whereas the *M*_n,GPC_ of PI_6.8_PS_17.3_AmPEO_1.9_ is shifted to a lower value at *M*_n,GPC_ = 31.2 kg mol^−1^ and for PI_26.1_PS_67.3_AmPEO_1.9_ to a higher value at *M*_n,GPC_ = 107.6 kg mol^−1^, which corresponds to their respective *M*_n,calc._ and *M*_n,NMR_ (cf. Figure 4, Table 4). The deviation towards higher molar mass in the *M*_n,GPC_ compared to the *M*_n,NMR_ values is attributed to the fact that the GPCs used polystyrene calibration standards. Furthermore, the GPC trace of EmPEO_1.9_ with *M*_n,GPC_ = 2.8 kg mol^−1^ shows a very narrow chain length distribution of *Đ* = 1.03. Thus, it was successfully characterized prior to linkage, showing a major advantage over the short PEO_z_ block control.

By linking the PEO_z_ block via EmPEO_z_, it prevents unwanted chain termination caused by the introduction of impurities, which has a considerable influence on *Đ*.

The result that the *Đ* values are close to one (*Đ* ≤ 1.08) indicates that the synthesized PI_x_PS_y_AmPEO_z_ polymer chains have nearly the same length, which means that there are almost no chain terminations during the polymerization process. This is consistent with the ^1^H-NMR results that the polymerization process is nearly quantitative and thus *M*_n,calc._ = *M*_n,NMR_. Therefore, this simplified polymerization procedure allows a fast and efficient polymer synthesis of tailored PI_x_PS_y_AmPEO_z_ BCPs in a controlled and reproducible way.

### 3.5. Thermal Analysis

The synthesized PI_x_PS_y_AmPEO_1.9_ polymers were analyzed by DSC (cf. Figure 5a). Besides the thermal induced phase transitions, these measurements were carried out in order to obtain relevant information regarding the microphase separation.

The occurrence of the three characteristic thermal phase transitions, i.e., the glass transition temperature (ϑ_g_) for the PI_x_ phase ϑ_g,PIx_ in the range of −68 to −60 °C, the ϑ_g,PSy_ for the PS_y_ phase in the range of 75 to 96 °C, as well as the melting point ϑ_mp,PEO1.9_ of the PEO_1.9_ (from AmPEO_1.9_) at ~50 °C (cf. Appendix A for DSC measurement of pure EmPEO_1.9_), evidence a microphase separation of the synthesized PI_x_PS_y_AmPEO_1.9_ BCPs (Table 5) [59]. The existence of ϑ_mp,PEO1.9_ indicates the phase separation of the PEO_1.9_ block from both nonpolar blocks. An exception is PI_26.1_PS_67.3_AmPEO_1.9_, where the ϑ_mp,PEO1.9_ is not detected. The reason for the absence of the ordered crystalline structure of the PEO_1.9_ chains in PI_26.1_PS_67.3_AmPEO_1.9_ can be attributed either to their low content of 1.7 wt.% or to the minimal contamination, which was detected in GPC measurement, so that the entire PEO_1.9_ block is amorphous. Moreover, the two ϑ_g_ and the ϑ_mp,PEO1.9_ are equal to those of the pure polymer blocks, and the absence of additional ϑ_g_ and ϑ_mp_ suggests the exclusion of any mixing of the individual blocks even at the phase boundaries. In consequence of the high tendency of phase separation (affected by the nearly monodisperse polymer chain lengths (*Đ* ≤ 1.08)), a correlation between the chain length of the individual polymer blocks and the respective ϑ_g_ is observed. An increase of the block length, which corresponds to a higher *M*_n_ of the corresponding polymer block, leads to a shift of the ϑ_g_ to higher temperature; for instance, an increase in *M*_n,PSy_ by ~10 kg mol^−1^ leads to a ϑ_g,PSy_ temperature increase of ~5 °C (cf. Table 5). Furthermore, the fact that there is no mixing between different polymer blocks indicates that a linear dependence of change of heat capacity Δ*C*_p_ with the weight fraction of the respective polymer block at ϑ_g_ is detected. For example, an increase in *M*_n,PIx_ of ~20 wt.% leads to a rise in the Δ*C*_p,PIx_ by ~0.12 J (g K)^−1^ for the ϑ_g,PIx_ (Table 5).

In Figure 5b, the decomposition temperature at a weight loss of 5% (ϑ_d5_) of EmPEO_1.9_ and the synthesized PI_x_PS_y_AmPEO_1.9_ polymers as determined by TGA measurements are shown. The ϑ_d5_ values are similar and independent of the polymer composition, with a slight trend of decreasing ϑ_d5_ with higher PI_x_ content and range from 322 to 334 °C. Therefore, they have a high thermal stability for the application as polymer electrolyte templates.

### 3.6. Morphological Characterization

Based on the previous results, it is evident that this polymerization route provides precise control over the chain length distribution respectively structure on a molecular level. In the following, it is investigated how this in combination with the controlled solvent casting process affects the morphology of the membrane. In particular, the self-assembly induced highly ordered microphase separation, which was already observed from DSC measurement, is crucial for the properties of the PI_x_PS_y_AmPEO_1.9_ polymers and therefore plays a key role in their application. The morphology of all synthesized PI_x_PS_y_AmPEO_1.9_ is determined using SAXS, SEM and TEM measurements. The results of above-mentioned morphological characterizations performed on the PI_x_PS_y_AmPEO_1.9_ membranes are shown in Figure 6, Figure 7, Figure 8 and Figure 9 and Table 6, which were controlled-cast from THF without annealing them later.

SAXS measurements were performed at room temperature on the as-cast membranes for all synthesized PI_x_PS_y_AmPEO_1.9_. Figure 6 and Table 6 show the results. All of the synthesized PI_x_PS_y_AmPEO_1.9_ specimens exhibited strong scattering, as seen in the 2D SAXS patterns in Figure 7, Figure 8 and Appendix A, and a clear first scattering peak (*q**), which was marked by a yellow filled circle in the SAXS curves in Figure 6, confirming the microphase separation [60,61].

**Table 6 polymers-15-02128-t006:** Summary of SAXS results determined at room temperature of the as-cast membranes of the different PI_x_PS_y_AmPEO_1.9_.

Polymer	*f*_PIx_ ^1^ /%	*f*_PSy_ ^1^ /%	*f*_PEO1.9_ ^1^ /%	*q** ^2^ /nm^−1^	Phase ^3^	*d* ^4^ /nm	*r*_cylinder_ ^5^ /nm	φ_cylinder_ ^6^ /%	Domain Size ^9^ /nm
PI_6.8_PS_17.3_AmPEO_1.9_	29	65	6	0.32	HEX	22.4	6.1	29 (PI_6.8_)	90
PI_14.6_PS_34.8_AmPEO_1.9_	32	65	3	0.20	HEX	35.7	10.4	34 (PI_14.6_)	122
PI_24.8_PS_25.0_AmPEO_1.9_	52	45	3	0.20	LAM	31.7	*t* ^7^ = 16.4	52 (PI_24.8_)_LAM_ ^8^	86
PI_35.1_PS_14.8_AmPEO_1.9_	71	26	3	0.19	HEX	39.2	10.9	31 (PS_14.8_AmPEO_1.9_)	63
PI_26.1_PS_67.3_AmPEO_1.9_	33	66	2	0.11	HEX	63.6	16.9	28 (PI_26.1_)	147

^1^ *f*_i_ = monomer volume fraction on basis of published homopolymer densities (*ρ*_PI_x__ = 0.830, *ρ*_PS_y__ = 0.969, *ρ*_PEO_z__ = 1.064 in g cm^−3^) [61] and calculated using *M*_n,NMR_ (cf. Table 3). Note that these densities were determined for 140 °C, which may cause deviations. ^2^ *q** is the position of the first scattering peak, determined by SAXS. ^3^ HEX = hexagonally close-packed cylindrical and LAM = lamella structure, determined by SAXS measurement. ^4^ *d* = average domain spacing. ^5^ *r*_cylinder_ = radius of cylinder determined by SAXS. ^6^ φ_cylinder_ = volume fraction of cylinder based on ratio of *r*_cylinder_ to *d*. ^7^ *t* = layer thickness determined by SAXS. ^8^ Volume fraction of lamella based on ratio of *t* to *d*. ^9^ Domain size = determined by SAXS.

In the SAXS curve of PI_6.8_PS_17.3_AmPEO_1.9_, in addition to the *q** = 0.32 nm^−1^, additional peaks were measured at a relative peak position at *q*/*q** of √1, √3, √4 and √7, indicating a HEX structure (cf. Figure 6) [18]. The same characteristic scattering peaks of a microphase-separated HEX morphology are detected for the membranes consisting of PI_14.6_PS_34.8_AmPEO_1.9_, PI_35.1_PS_14.8_AmPEO_1.9_ and PI_26.1_PS_67.3_AmPEO_1.9_, as shown in Figure 6 [18]. However, all *q* values of these polymers are shifted in comparison to PI_6.8_PS_17.3_AmPEO_1.9_ to lower *q* values, whereby PI_26.1_PS_67.3_AmPEO_1.9_ has the lowest *q** value with 0.11 nm^-1^. In the case of PI_24.8_PS_25.0_AmPEO_1.9_, the *q** value = 0.20 nm^−1^ is the same as for PI_14.6_PS_34.8_AmPEO_1.9_ (0.20 nm^−1^) and PI_35.1_PS_14.8_AmPEO_1.9_ (0.19 nm^−1^). In addition to the *q** peak, further peaks at a relative position of *q*/*q** 2 and 3 are clearly observed for PI_24.8_PS_25.0_AmPEO_1.9_, indicating a LAM morphology (cf. Figure 6) [18,60].

The *q** values were used to calculate the respective average domain spacing (*d*) for all PI_x_PS_y_AmPEO_1.9_; this means for a HEX structure, the distance is from cylinder to adjacent cylinder, and for a LAM structure, from center to the next center, which is listed in Table 6. In addition, from SAXS measurement of PI_24.8_PS_25.0_AmPEO_1.9_ the layer thickness (*t*) and for samples with the HEX morphology, the cylinder radius (*r*_cylinder_) were determined and summarized in Table 6. Details concerning the fits to the radially averaged SAXS data in Figure 6 and the values in Table 6 are given in the supporting information.

The extra hump in the SAXS curve from PI_6.8_PS_17.3_AmPEO_1.9_ at low *q* value = 0.16 nm^−1^, which therefore precedes the *q** value = 0.32 nm^−1^ and was not fitted, comes most probably from heterogeneities in the structure on length scales larger than the unit cell. The reason that only the one peak is seen and the others are suppressed lies in the ratio of *r*_cylinder_ and *d* because in this combination, the peaks fall on the minima and are suppressed.

For all investigated polymers, the PEO_1.9_ block is with a volume fraction (φ) of 2–6% too small to be detected by SAXS measurements. Thus, it was not necessary to include the PEO_1.9_ block in the fit, as from DSC measurements it is known to be phase-separated from the nonpolar blocks, so a simplified diblock model was used.

For all samples, the fit used to determine the morphology matches very well to the measured SAXS curve shapes (cf. plotted black line in the SAXS curves in Figure 6). Therefore, the calculated relative peak positions at *q*/*q** (cf. black squares for HEX and black stars for LAM structure in Figure 6) fit almost perfectly to the maxima and minima in the respective obtained SAXS curves. The model used for the fitting is either using homogenous cylinder form factor for the HEX or platelets form factor for the LAM phases. Even with this rather simple model, a precise fit of the radially averaged scattering data was achieved. The φ for different blocks were calculated from the ratio of *r*_cylinder_ or *t* to the fitted *d* and are listed in Table 6.

The *d* values calculated from measured *q** correlate with the total *M*_n,NMR_ of the PI_x_PS_y_AmPEO_1.9_. This means that PI_6.8_PS_17.3_AmPEO_1.9_ possesses not only the smallest structure size with *d* = 22.4 nm, but also the fact that it is about half of the size compared to PI_14.6_PS_34.8_AmPEO_1.9_ with *d* = 35.7 nm, PI_24.8_PS_25.0_AmPEO_1.9_ with *d* = 31.7 nm and PI_35.1_PS_14.8_AmPEO_1.9_ with *d* = 39.2 nm. The same behavior, but this time reversed, is seen for PI_26.1_PS_67.3_AmPEO_1.9_, which not only has the largest structure size with *d* = 63.6 nm, but also its size is in comparison to PI_14.6_PS_34.8_AmPEO_1.9_, PI_24.8_PS_25.0_AmPEO_1.9_ and PI_35.1_PS_14.8_AmPEO_1.9_ about two times larger. Therefore, the structure size ratios of the different PI_x_PS_y_AmPEO_1.9_ to each other are in accurate accordance with their respective *M*_n,NMR_ (cf. Figure 4, Table 4). Also, the measured *r*_cylinder_ have the same relationship to each other as the previously described *d* (cf. Table 6).

Using the *M*_n,NMR_ of the individual polymer blocks (cf. Table 3) and on basis of published homopolymer densities (*ρ*_PIx_ = 0.830, *ρ*_PSy_ = 0.969, *ρ*_PEOz_ = 1.064 in g cm^−3^_,_ note that these densities were determined for 140 °C, which may cause slight deviations) [61], the monomer volume fractions (*f*_i_) of the PI_x_ block (*f*_PIx_), the PS_y_ block (*f*_PSy_) and the PEO_1.9_ block (*f*_PEO1.9_) were calculated and listed in Table 6. For PI_6.8_PS_17.3_AmPEO_1.9_, PI_14.6_PS_34.8_AmPEO_1.9_ and PI_26.1_PS_67.3_AmPEO_1.9_, the volume ratio of the PI_x_ block to the PS_y_AmPEO_1.9_ block always remains the same and was in the range of ~29–33% for *f*_PIx_ and ~68–71% for *f*_PSyAmPEO1.9_. The volume fraction of the cylinder (φ_cylinder_) was equal for all with ~28–34%, which matches very well with the *f*_PIx_. Accordingly, it was obvious that they have the same HEX structure. Consequently, the cylinder from the HEX structure corresponds to the PI_x_ block, because the *r*_cylinder_ ratio of the different polymers, i.e., PI_6.8_PS_17.3_AmPEO_1.9_, PI_14.6_PS_34.8_AmPEO_1.9_ and PI_26.1_PS_67.3_AmPEO_1.9_, agrees exactly with the respective *M*_n,NMR,PIx_ ratio. As a result, the HEX structures differ from each other only in terms of size.

The volume ratio for PI_35.1_PS_14.8_AmPEO_1.9_ with *f*_PI35.1_ = 71% and *f*_PS14.8AmPEO1.9_ = 29% is exactly the opposite compared to the previously discussed PI_x_PS_y_AmPEO_1.9_. The determined φ_cylinder_ with 31% matches quite closely with the *f*_PS14.8AmPEO1.9_. Consequently, the detected HEX structure fits accurately; however, due to the inverted volume fractions, a reverse phase with PS_14.8_AmPEO_1.9_ cylinders exists. Hence, *r*_cylinder_ with a value of 10.9 nm corresponds to the *M*_n,NMR_ of PS_14.8_ and AmPEO_1.9_ block and is the same size as *r*_cylinder_ of PI_14.6_PS_34.8_AmPEO_1.9_ as well as its *M*_n,NMR_ of the PI_14.6_ block. This fact confirms that PI_35.1_PS_14.8_AmPEO_1.9_ has the reverse phase of the HEX structure of the PI_14.6_PS_34.8_AmPEO_1.9_.

The measured LAM morphology for PI_24.8_PS_25.0_AmPEO_1.9_ with the volume ratio of *f*_PI24.8_ = 52% and *f*_PS25.0AmPEO1.9_ = 48% fits exactly with the determined almost equal volume fraction of lamella (φ_LAM_) of the PI_24.8_ block with φ_LAM_ = 52% and 48% for the PS_25.0_AmPEO_1.9_ block.

Indeed, as expected from the PI_x_PS_y_ phase diagram, the nearly symmetric PI_24.8_PS_25.0_AmPEO_1.9_ forms a LAM, while all other, more asymmetric PI_x_PS_y_AmPEO_1.9_ show a HEX morphology [18,24].

In order to visualize the previously described structures more clearly, corresponding 3D drawings were plotted based on the parameters obtained from the SAXS measurements with the assumption that the PEO_1.9_ block is fully phase-separated (cf. 3D drawings on the right site in Figure 6).

The obtained SAXS results match almost perfectly for all PI_x_PS_y_AmPEO_1.9_ with their respective polymer composition determined from NMR, thermal analysis and GPC measurements, and in particular, when they are compared with each other.

For all PI_x_PS_y_AmPEO_1.9_, the SAXS measurement results indicate a high degree of long-range order in the structure due to the strong scattering in the respective 2D SAXS pattern, but also due to the fact that the curve shape shows a number of clear peaks that agree very well with the fit as well as with the calculated peak positions. For instance, in PI_24.8_PS_25.0_AmPEO_1.9_ the presence of 2*q** and 3*q** suggests that the LAM morphology is present with a highly long-range arrangement [60,62].

In order to determine a more detailed, local long-range order as well as the orientation of the structure, SEM measurements were carried out. In particular, these two structural parameters of the membrane are crucial for the application of BCP as a structure-giving polymer matrix in electrolyte. In addition, the structure determined from the SAXS results is also checked. For this purpose, exemplary SEM measurements were performed on unstained ultra-thin sections of about 50 to 100 nm thickness from PI_14.6_PS_34.8_AmPEO_1.9_ for the HEX structure and from PI_24.8_PS_25.0_AmPEO_1.9_ for the LAM structure. In Appendix A, the procedure of sample preparation by cryo-ultramicrotomy as well as sample placement onto the grid is schematically shown.

Results of PI_14.6_PS_34.8_AmPEO_1.9_ are shown in Figure 7; the ones for PI_24.8_PS_25.0_AmPEO_1.9_ are displayed in Figure 8. In (a), an overview image of several µm^2^ in size is shown. The marked area in a) was enlarged as inset in b) to make the structure visible in more detail (i.e., periodic lamellae or hexagonal cylinder arrangement). In (c), the fast Fourier transformation (FFT) of the area in (a) in case of the LAM and of the inlet (b) in case of the HEX structure is added to indicate the average orientation of the structure. The good contrast of the sample is achieved through the relatively low accelerating voltage of 30 kV used for STEM-imaging in the SEM. Hence, samples can be measured without staining. Additionally, in both figures under (d), the corresponding SAXS pattern is shown to compare it with the FFT spectrum.

For PI_14.6_PS_34.8_AmPEO_1.9_, a long-range and highly ordered HEX structure of cylinders is clearly visible in the overview image of Figure 7a, respectively, in the enlarged inset in (b) and corresponds with the SAXS result. In particular, it is remarkable that the hexagonal cylinder pattern of the HEX structure is clearly visible and almost identical in the FFT spectrum in (c) as well as in the SAXS pattern in (d). This means that the cylinders have the same arrangement and distance to each other in the macroscopic sample volume (2 mm sample diameter in the SAXS capillaries) and in the microscopic sample area in the SEM image and are aligned straight through the entire volume. Furthermore, from the SEM image, it is evident that for the cast film HEX structure, the cylinder axes are located in plane.

In case of PI_24.8_PS_25.0_AmPEO_1.9_ in the large overview image in Figure 8a, respectively, in the inset in b), a long-range and highly ordered LAM structure is clearly evident, which is also consistent with the SAXS result and corroborated by the 2D FFT result. The obtained 2D FFT spectrum reveals two “beam-like shapes” according to the LAM structure going vertical through the sample (cf. Figure 8c). Also, the SAXS pattern in Figure 8d shows the two “beam-like shapes” which differ from those in the FFT nearly only by the fact that they are slightly tilted. Most likely this is due to the placement of the sample into the SAXS capillary. Considering that the FFT spectrum and the SAXS pattern look almost the same, the lamellae are microscopically as well as macroscopically equally oriented and have the same distance to each other.

Moreover, a TEM image of PI_24.8_PS_25.0_AmPEO_1.9_ of the unstained membrane was obtained in STEM mode. Usually, stained samples are measured [18,24,63,64]. The TEM result shown in Figure 9 displays the same LAM structure as already seen in SEM.

As a result, PI_14.6_PS_34.8_AmPEO_1.9_ and PI_24.8_PS_25.0_AmPEO_1.9_ possess a nearly perfectly ordered HEX respectively LAM structure, at least over the total SEM and TEM measurement µm size area as well as the 2 mm thick sample volume for the SAXS measurement. Therefore, they show an extraordinary long-range orientation. The strongly distinct microphase separation, most likely caused by the very narrow polymer chain length distribution, in combination with the controlled solution casting process ensuring enough time for self-assembly, leads to these exceptional morphological properties. These properties are crucial for an excellent polymer electrolyte matrix, because the conductive pathways and finally the Li^+^ transport should be aligned almost optimally to each other over an extremely long range inside the BCP membrane.

Furthermore, particularly in case of PI_24.8_PS_25.0_AmPEO_1.9_, horizontal cutting artifacts can be recognized in the sample overview image as brighter and darker waves. This is a common cutting artifact of cryo-ultramicrotomy, which results from compression of the section during sectioning and extends perpendicular to the cutting direction [65]. As the cutting direction was perpendicular to the membrane and a ribbon of sections could be imaged on the grid by SEM, it is possible to determine the orientation of the lamellae within the membrane over a larger sample surface area and at several sample positions (cf. Appendix A). The lamellae are arranged from top to bottom, i.e., vertically across the sample. Compression artefacts are perpendicular to the lamellae, indicating that the cutting direction across the membrane is the same as the orientation of the lamellae in the BCP. In other words, the lamellae are continuous aligned orthogonally to the polymer surface and therefore also at the potential use in electrolytes as a structure-giving BCP matrix to the electrode interface, i.e., connecting both electrodes to each other. It has already been shown in our patent that these BCPs can be used as a structure-giving BCP matrix in electrolytes to obtain very high ionic conductivities [43]. Such electrolytes provide comparable, and in some cases even better, ionic conductivities than those reported by Dörr and Pelz et al. [10,11,37].

## 4. Conclusions

Herein, we introduced a convergent synthesis method based on the modular principle, which ensures the access to well-defined PI_x_PS_y_PEO_z_ linear triblock copolymers with consistently the same very short and precise PEO_z_ block. For this purpose, a prefabricated and commercially available mPEO_z_ block is used, which is selectively functionalized with an epoxy end group to EmPEO_z_. The EmPEO_z_ block is covalently attached to the PI_x_PS_y_^−^ anion synthesized by living anionic polymerization and thus terminated to the corresponding BCP. Thereby, by utilizing the O-Li ion pair formation between the epoxide group of the EmPEO_z_ and the living anion, it is ensured that only a single ring-opening reaction occurs. Therefore, the handling of EO gas monomers during the BCP polymerization can be avoided.

The systematic variation of the block length reveals two major and independent influences of the polymer structure. By varying the *M*_n,PIx_/*M*_n,PSy_ at a constant *M*_n,total_, the morphology could be precisely controlled. Alternatively, by changing the *M*_n,total_ at the same *M*_n,PIx_/*M*_n,PSy_, only the PEO_z_ block fraction could be altered and adjusted, while keeping the morphology constant.

This simplified and reproducible one-pot polymerization, with 100% reaction efficiency, obtains highly ordered PI_x_PS_y_AmPEO_z_ BCPs whose morphology can be largely controlled within the polymer membrane on microscopic and macroscopic levels. Largely controlled means at the microscopic or molecular level that the polymer chain lengths can be precisely selected, are nearly monodisperse and the respective BCP possesses a very high degree of phase separation. Moreover, the highly uniform polymer blocks combined with the controlled solution casting process lead to PI_x_PS_y_AmPEO_z_ membranes with a very high and exceptionally long-range ordered structure up to the mm scale.

Overall, due to the always constant PEO_z_ block size, the high control over the order and orientation of the morphology as well as the PEO_z_ block fraction, these BCPs are suitable for subsequent targeted investigation of their influence as a structure-giving matrix in corresponding solid polymer electrolytes with respect to Li^+^ transport up to the macroscale level.

## Data Availability

The data presented in this study are available from the corresponding authors on request.

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
