# Peer review of "Improved Route to Linear Triblock Copolymers by Coupling with Glycidyl Ether-Activated Poly(ethylene oxide) Chains"

_polymers, 2023, doi:10.3390/polym15092128_

Round 1

Reviewer 1 Report

It is a well-written manuscript with a detailed description of the synthesis and characterization of the block copolymer. However, the reviewer did not find convincing results/explanations of the characteristics of polymer to be used in Li+ ion batteries. Therefore, authors should do experiments such as conductivity or electrochemical measurements using the block polymer.

Author Response

Comments:

It is a well-written manuscript with a detailed description of the synthesis and characterization of the block copolymer. However, the reviewer did not find convincing results/explanations of the characteristics of polymer to be used in Li+ ion batteries. Therefore, authors should do experiments such as conductivity or electrochemical measurements using the block polymer.

Respond to reviewer:

We would like to thank the reviewer for his overall positive feedback and comments, including the remark about missing conductivity measurement.

We have filed a now publicly available patent (PCT/WO 2022/008615 A1) about the synthetic method that is described and discussed in this publication. Therefore, we are able to report about it in this publication. In the manuscript, the patent is mentioned in the abstract (lines 41 â€‘ 43), introduction (lines 145 and 146), and conclusion (lines 838 â€‘ 842), with appropriate reference [43].

This patent has arisen after the publications of Dörr and Pelz et al. and only contains ionic conductivity data of electrolytes in which the BCPs were prepared by the synthesis method presented here. The ionic conductivity values published in the patent are comparable and in some cases even higher than those reported by Dörr and Pelz et al.. From our point of view, this fact should sufficiently demonstrate that BCPs obtained by the synthesis route described here are usable as a structure-giving matrix in electrolytes to achieve high ionic conductivities.

Of course, it would be nice to show the ionic conductivity values from the patent again. However, we believe that it would not be appropriate to repeat a diagram that has already been published. Nevertheless, if there is an explicit wish to include these conductivity data in this manuscript, naturally we are prepared to do so, but we have to mark this as a citation accordingly.

Reviewer 2 Report

1. It was mentioned in Abstract that the synthetic route that could adjust the length of PIx and PSy was developed in order to systematically discuss the influence of the block length and related morphology of PIx and PSy on ionic conductivity. It was suggested to discuss and analyze the conductivity of each polymer.

2. The figures should be mentioned in the manuscript in order. For example, Figure 4 should be mentioned before Figure 5, in Lines 186-187. Please check the manuscript carefully and revise it.

3. The caption of the Figures and the Tables was prolix, while the description of the analysis of the Figure and the Table were so brevity in the manuscript. Please check manuscript and the modify the relevant content reasonably.

4. In the manuscript, the exact number of decimal places in the data is not uniform, such as “48.0, 120 in Table 1, 0.13 - 6 nm-1” in Section 2.9 and etc. Please check the manuscript carefully.

5. The citation format of the Figure and the Table in the text is not uniform, such as cf. Figure S1 and Figure 2 b”. Please standardize the format.

6. There were some formatting errors in this manuscript. “Scheme 2” should be replaced by “Scheme 2.” in Line 132.

7. In References, “TiO2” should be corrected as “TiO2” in Line 916. And there were errors in the formats of [13], [27] and [42]. Please revise after checking carefully them.

8. The authors should add references in the last five years in order to show the research progress on the synthesis route of this kind of polymers and prove the significance and innovation of this work.

9. As for the Supplementary Materials, the Figure S10 was repeated with Figures S5, S6, S7, S8 and S9, so please deleted them appropriately.

Author Response

This work introduced a convergent synthesis method based on the modular principle, which ensures the access to well-defined PIxPSyPEOz linear triblock copolymers. In general, this work was well conducted and organized. The topic is also interesting, with a wide audience. Therefore, I recommend the acceptance after the following revisions will be made. I list my detailed comments on this manuscript below.

Comment:

  1. It was mentioned in Abstract that the synthetic route that could adjust the length of PIx and PSy was developed in order to systematically discuss the influence of the block length and related morphology of PIx and PSy on ionic conductivity. It was suggested to discuss and analyze the conductivity of each polymer.

Respond to reviewer:

We thank the reviewer for his positive feedback and constructive criticism about the missing discussion of ionic conductivities depending on the polymer electrolyte composition.

However, the desired discussion of the ionic conductivity in dependence of the different BCP types is beyond the scope of this publication. Furthermore, it would make the present manuscript much too long.

Moreover, in our now publicly available patent (PCT/WO 2022/008615 A1) about the synthetic route described and discussed in this publication, ionic conductivity data have been published. In the manuscript, the patent is mentioned in the abstract (lines 41 â€‘ 43), introduction (lines 145 and 146), and conclusion (lines 838 â€‘ 842), with appropriate reference [43].

This patent has arisen after the publications of Dörr and Pelz et al. and contains only ionic conductivity data of electrolytes in which the BCPs were prepared by the synthesis method presented here. The ionic conductivity values published in the patent are comparable and in some cases even higher than those reported by Dörr and Pelz et al.. From our point of view, this fact should sufficiently demonstrate that BCPs obtained by the synthesis route described here are usable as a structure-giving matrix in electrolytes to achieve high ionic conductivities.

However, the interpretation of the ionic transport mechanism that leads to these very high conductivities will be discussed in another forthcoming publication. For this purpose, the recent conductivity values required are already available and do not differ significantly from those in the previously mentioned publications.

Comment:

  1. The figures should be mentioned in the manuscript in order. For example, Figure 4 should be mentioned before Figure 5, in Lines 186 â€‘ 187. Please check the manuscript carefully and revise it.

Respond to reviewer:

For his comment, we thank the reviewer. The order of the figures is now updated accordingly to the remark.

Comment:

  1. The caption of the Figures and the Tables was prolix, while the description of the analysis of the Figure and the Table were so brevity in the manuscript. Please check manuscript and the modify the relevant content reasonably.

Respond to reviewer:

Thanks to the reviewer for his comment. Some of the labels have been shortened, according to the comment. However, for some of them it is necessary to give such a precise indication, so that they are complete and self-explanatory in combination with the figure/table/scheme (e.g. Figure 6).

Comment:

  1. In the manuscript, the exact number of decimal places in the data is not uniform, such as “48.0, 120” in Table 1, “0.13 – 6 nm‑1” in Section 2.9 and etc. Please check the manuscript carefully.

Respond to reviewer:

We would like to thank the reviewer for this correct remark. Accordingly, equal numbers of decimal places were used for all data within the manuscript.

Comment:

  1. The citation format of the Figure and the Table in the text is not uniform, such as “cf. Figure S1” and “Figure 2 b”. Please standardize the format.

Respond to reviewer:

We thank the reviewer for his comment. According to the note, the space between the number and the letter in the figures, tables and schemes have been uniformly removed (e.g. Figure 5a).

Comment:

  1. There were some formatting errors in this manuscript. “Scheme 2” should be replaced by “Scheme 2.” in Line 132.

Respond to reviewer:

Thanks to the reviewer for his comment. According to the note, the label Scheme 2 was corrected to Scheme 2.. In addition, this was checked for all other figure, table and scheme labels and corrected as necessary.

Comment:

  1. In References, “TiO2” should be corrected as “TiO2” in Line 916. And there were errors in the formats of [13], [27] and [42]. Please revise after checking carefully them.

Respond to reviewer:

We would like to thank the reviewer for this correct remark. Accordingly, the specified as well as all other references have been corrected.

Comment:

  1. The authors should add references in the last five years in order to show the research progress on the synthesis route of this kind of polymers and prove the significance and innovation of this work.

Respond to reviewer:

Many thanks to the reviewer for the corrected and useful hint. According to the hint, there were inserted literatures from the years 2019 â€‘ 2023, with the updated reference numbers [25 – 27, 35, 36, 41, 50].

Comment:

  1. As for the Supplementary Materials, the Figure S10 was repeated with Figures S5, S6, S7, S8 and S9, so please deleted them appropriately.

Respond to reviewer:

Also, we would like to thank the reviewer for his comment. In figure S10. a section of the 1H‑NMR measurements from figure S5. to figure S9. is shown for comparison of the characteristic signals. In our opinion, figures S5. to S9. are necessary for completeness and to show that there are no other signals in the 1H‑NMR spectra.

Reviewer 3 Report

The authors present a well written research regarding the "Improved Route to Linear Triblock Copolymers by Coupling with Glycidyl Ether-Activated Poly(Ethylene Oxide) Chains."

In my opinion, this work is very interesting and well-structured. The experiments and conclusions are clear and well supported by the results. The study is thorough and well structured.

The authors should just address some typos and minor English grammatical errors before pubblication.

Author Response

The authors present a well written research regarding the "Improved Route to Linear Triblock Copolymers by Coupling with Glycidyl Ether-Activated Poly(Ethylene Oxide) Chains."

In my opinion, this work is very interesting and well-structured. The experiments and conclusions are clear and well supported by the results. The study is thorough and well structured.

The authors should just address some typos and minor English grammatical errors before publication.

Respond to reviewer:

We thank the reviewer for his comments and very positive feedback. Corresponding typing and legal spelling errors have been improved.